# ACHIEVING MORPHOLOGICAL AGREEMENT WITH CONCORDE

## ABSTRACT

Neural conversational models are widely used in applications like personal assistants and chat bots. These models seem to give better performance when operating on word level. However, for fusion languages like French, Russian and Polish, vocabulary size sometimes become infeasible since most of the words have lots of word forms. To reduce vocabulary size we propose a new pipeline for building conversational models: first generate words in standard form and then transform them into a grammatically correct sentence. For this task we propose a neural network architecture that efficiently employs correspondence between standardised and target words and significantly outperforms character-level models while being 2x faster in training and 20% faster at evaluation. The proposed pipeline gives better performance than character-level conversational models according to assessor testing.

## 1 INTRODUCTION

Neural conversational models Vinyals & Le (2015) are used in a large number of applications: from technical support and chat bots to personal assistants. While being a powerful framework, they often suffer from high computational costs.

The main computational and memory bottleneck occurs at the vocabulary part of the model. Vocabulary is used to map a sequence of input tokens to embedding vectors: one embedding vector is stored for each word in vocabulary.

English is de-facto a standard language for training conversational models, mostly for a large number of speakers and simple grammar. In english, words usually have only a few word forms. For example, verbs may occur in present and past tenses, nouns can have singular and plural forms.

For many other languages, however, some words may have tens of word forms. This is the case for Polish, Russian, French and many other languages. For these languages storing all forms of frequent words in a vocabulary significantly increase computational costs.

To reduce vocabulary size, we propose to normalize input and output sentences by putting them into a standard form. Generated texts can then be converted into grammatically correct ones by solving morphological agreement task. This can be efficiently done by a model proposed in this work.

Our contribution is two-fold:

- We propose a neural network architecture for performing morphological agreement in fusion languages such as French, Polish and Russian (Section 2).

- We introduce a new approach to building conversational models: generating normalized text and then performing morphological agreement with proposed model (Section 3);

## 2 CONCORDE

In this section we propose a neural network architecture for solving morphological agreement problem.

We start by formally defining the morphological agreement task. Consider a grammatically correct sentence with words $[a_1, a_2, \ldots, a_K]$. Let $\mathcal{S}(a)$ be a function that maps any word to its standard form. For example, $\mathcal{S}(\text{"went"}) = \text{"go"}$. Goal of morphological agreement task is to learn a mapping from normalized sentence $[\mathcal{S}(a_1), \mathcal{S}(a_2), \ldots, \mathcal{S}(a_K)] = [a_1^n, a_2^n, \ldots, a_K^n]$ to initial sentence $[a_1, a_2, \ldots, a_K]$. Interestingly, reverse mapping may be performed for each word independently using specialized dictionaries. Original task, however, needs to consider dependencies between words in sequence in order to output a coherent text.

An important property of this task is that the number of words, their order and meaning are explicitly contained in input sequence. To employ this knowledge, we propose a specific neural network architecture illustrated in Figure 1. Network operates as follows: first all normalized words are embedded using the same character-level LSTM encoder. The goal of the next step is to incorporate global information about other words to embedding of each word. To do so we pass word embedding sequence through a bidirectional LSTM Graves et al. (2013). This allows new embeddings to get information from all other words: information about previous words is brought by forward LSTM and information about subsequent words is taken from backward LSTM. Finally, new embeddings are decoded with character-level LSTM decoders. At this stage we also added attention Bahdanau et al. (2015) over input characters of corresponding words for better performance.

High-level overview of this model resembles sequence-to-sequence Sutskever et al. (2014) network: model learns to map input characters to output characters for each word using encoder-decoder scheme. The difference is that bidirectional neural network is used to distribute information between different words.

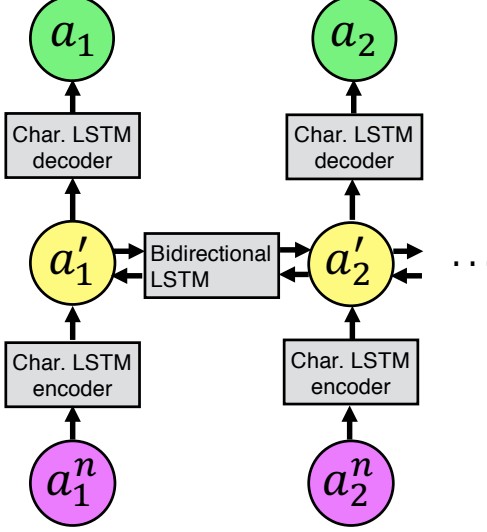

Figure 1: Concorde model.

Unlike simple character-level sequence-to-sequence model, our architecture allows for much faster evaluation, since encoding and decoding phases can be done in parallel for each word. Also, one of the main advantages of our approach is that information paths from inputs to outputs are much shorter which leads to slower performance degradation as input length increases (see Section 5.3).

## 3  Q-CONCORDE: NEURAL CONVERSATION

As discussed above, we propose a two stage approach to build a neural conversational model: first generate normalized answer using normalized question and then apply Concorde model to obtain grammatically correct response. In this section we discuss a modification of Concorde model for conditioning its output on question's morphological features.

A modification of Concorde model that we call Q-Concorde uses two sources of input: question and normalized answer. Question is first embedded into a single vector with character-level RNN. This

vector may carry important morphological information such as time, case and plurality of questions. Question embedding is then mixed with answer embeddings using linear mapping. The final model is shown in Figure 3.

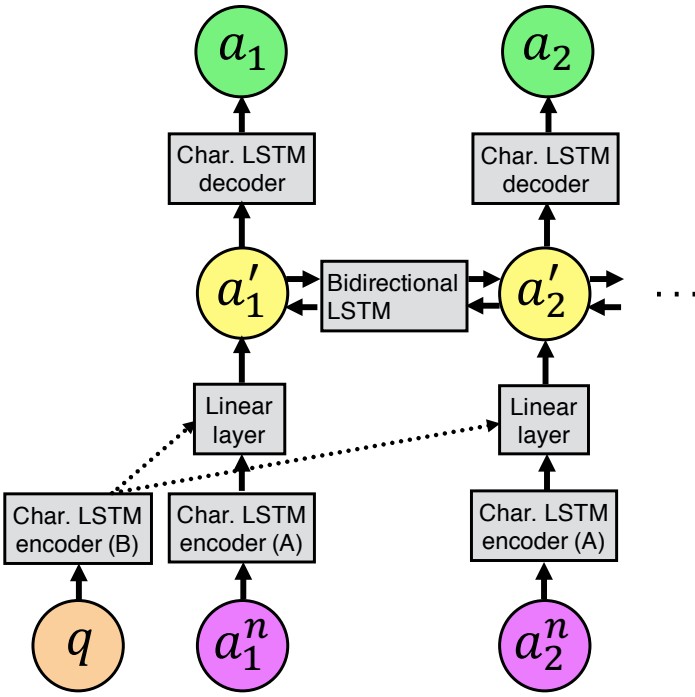

Figure 2: Q-Concorde model.

## 4 RELATED WORK

Most frequently used models for sequence to sequence mapping rely on generating embedding vector that contains all information about input sequence. Reconstruction solely from this vector usually results in worse performance as length of the output sequence increases. Attention (Xu et al. (2015), Luong et al. (2015),Bahdanau et al. (2015)) partially fixes this problem, though information bottleneck of embedding vector is still high.

Encoder-decoder models have mostly been applied to tasks like speech recognition (Graves & Schmidhuber (2005), Graves et al. (2013), Graves & Jaitly (2014)), machine translation (Bahdanau et al. (2015), Sutskever et al. (2014), Cho et al. (2014)) and neural conversational models (Vinyals & Le (2015), Shang et al. (2015)).

Some works have tried to perform decomposition of input sequence to obtain shorter information paths. In (Johansen et al. (2016)) input is first processed character-wise and then embeddings that correspond to word endings are used for sequence-to-sequence model. This modification makes input-output information paths shorter which leads to better performance.

Word inflection is a most similar task to ours, but in this task model is asked to generate a specific word form while we want our model to automatically select desired word forms. Durrett & DeNero (2013) proposed a supervised approach to predicting the set of all word forms by generating transformation rules from known inflection tables. They also propose to use Conditional Random Fields for unseen base forms.

Some authors have also tried to apply neural networks for this problem. Aharoni et al. (2016) and Faruqui et al. (2015) propose to use bidirectional LSTM to encode the word. Then Faruqui et al. (2015) uses different decoders for different word forms, while Aharoni et al. (2016) suggests to have one decoder and to attach morphological features to its input.

Besides recurrent networks, there has been an attempt to use convolutional networks. Ostling (2016) based his work on Faruqui et al. (2015) and proposed to first pass raw data through convolutional layers and then to pass them through recurrent encoder. Kim et al. (2016) uses character-level encoder and word-level decoder.

There has also been some work on achieving open vocabulary. Luong & Manning (2016) suggests to use character-wise encoder to generate embeddings of out-of-vocabulary words

## 5 EXPERIMENTS

We evaluate our model in two steps. First we compare performance of Concorde and character-level models in three languages: French, Polish, and Russian. We then evaluate Q-Concorde model on Q&A task.

### 5.1 SETUP

To construct training sets for French and Polish languages we use normalization vocabularies[1]. For normalization of Russian language we use pymorphy2 Korobov (2015) library. We leave first 20 words from each sentence to reduce computational costs. Concorde and Q-Concorde models consist of 2-layer LSTM encoder and decoder with hidden size 512.

We compare our model to three baselines: unigram charRNN, bigram charRNN and hierarchical model. Unigram and bigram models are standard sequence-to-sequence models with attention Luong et al. (2015) that operate with characters or pairs of characters as tokens. We use 2-layer LSTM as an encoder. Decoder consists of 2-layer LSTM followed by attention layer and another recurrent layer. The third baseline is a hierarchical model motivated by Johansen et al. (2016): we first embed each word using recurrent encoder and then compute sentence embedding by running word-level encoder on these embeddings. For baselines we use layer size of 768 which results in a comparable number of parameters for all models.

We train models with Adam Kingma & Ba (2014) optimizer in batch size 16 with learning rate 0.0002 that halves after each 50k updates. We terminate training after 300k updates which is enough for all models to converge.

### 5.2 OPENSUBTITLES

To evaluate our model we used French, Russian an Polish corpuses from OpenSubtitles[2] database. We performed morphological agreement for each subtitle line independently. We estimated potential vocabulary size reduction from normalization by selecting words that appeared more than 10 times in first 10M examples. This lead to 2.5 times reduction for Polish language, 2.4 for Russian, and 1.8 for French.

We evaluated our model in two metrics: word and sentence accuracies. Word accuracy shows a fraction of words that were correctly generated. Sentence accuracy corresponds to the fraction of sentences that were transformed without any mistake. Results are reported in Table 1. From four models that we compared, our model gave the best performance among all datasets, while second best model was hierarchical model.

We inspected our model to show some examples where it was able to infer plural form and gender for unseen words (Table 2). For Russian language we found out that the model was able to learn some rare rules like changing the letter «я» to «й» when going to plural form in some words: «один заяц», «два зайца» (one rabbit, two rabbits).

We can also see that our model can infer gender from words. To show that we chose feminine, masculine and neuter words and asked the model to perform agreement with word "one". This word changes its spelling for different genders in French, Polish, and Russian. Results presented in Table 3 suggest that model can indeed correctly solve this task by setting correct gender to the numeral.

---

[1] http://www.lexiconista.com/datasets/lemmatization/
[2] http://opus.lingfil.uu.se/OpenSubtitles2016.php

Table 1: Performance of our model for different languages. Results are reported in word accuracy (**w**) and sentence accuracy (**s**).

|  |  | **CharRNN** | | **Bigrams** | | **Hierarchical** | | **Concorde** | |
|---|---|---|---|---|---|---|---|---|---|
|  |  | **w** | **s** | **w** | **s** | **w** | **s** | **w** | **s** |
| **French** | **train** | 84.53 | 50.83 | 53.47 | 39.36 | 85.34 | 51.54 | **89.18** | **58.34** |
|  | **test** | 83.10 | 50.50 | 53.08 | 40.35 | 83.03 | 50.52 | **87.90** | **56.86** |
| **Russian** | **train** | 73.89 | 42.60 | 48.07 | 35.35 | 75.67 | 44.16 | **83.95** | **51.94** |
|  | **test** | 75.88 | 44.97 | 49.75 | 37.29 | 77.73 | 46.26 | **85.18** | **53.97** |
| **Polish** | **train** | 70.40 | 38.51 | 45.58 | 31.16 | 70.50 | 38.30 | **79.51** | **45.10** |
|  | **test** | 70.41 | 38.02 | 44.07 | 31.10 | 70.71 | 38.31 | **79.76** | **44.93** |

Table 2: Model inspection: single and plural forms.

|  | **Single** | | **Plural** | |
|---|---|---|---|---|
|  | **Normalized** | **Output** | **Normalized** | **Output** |
| **French** | un château | Un château | deux château | Deux châteaux |
| **Russian** | один заяц | Один заяц | два заяц | Два зайца |
| **Polish** | jeden ołówek | Jeden ołówek | dwa ołówek | Dwa ołówki |

Finally in Table 4 we show results on full sentences. Interestingly, on quite a complex Russian example our model was able to perform agreement. To select the correct form of word «соседнем» (neighbouring), network had to use multiple markers from different parts of a sentence: gender from «подъезд» (entrance) and case from «в» (in).

## 5.3 PERFORMANCE ON LONG SEQUENCES

As a motivation for our model we argued that making shorter input-output paths may reduce information load of the embedding vector. To check this hypothesis we computed average sentence accuracy for different input lengths and reported results in Figure 3.

We can clearly see that all baseline models perform worse as the input length increases. However, this is not the case for our model — while character-level models perform with almost 0% accuracy when input is 100 characters long, our model still gives similar performance as for short sentences.

This result can be explained by the way in which models use embedding vectors. Baseline models have to share embedding capacity between all words in a sentence. Our model, however, has a separate embedding for each word and does not require the whole sentence to be squeezed into one vector.

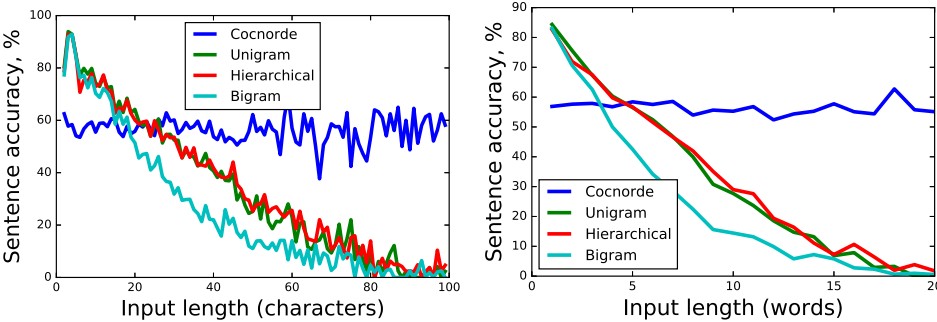

Figure 3: Test accuracy on French corpus for different input lengths.

It is also clear that character-level models perform better for short sequences (about 33% of the test set). This may be the case since the capacity of the embedding vector is not fully used for them.

Table 3: Model inspection: gender. **f.** is Feminine, **m.** is Masculine and **n.** is Neuter.

|  |  | **Normalized** | **Output** |
|---|---|---|---|
| **French** | **f.** | un personne | Une personne |
|  | **m.** | un témoin | Un témoin |
| **Russian** | **f.** | один тарелка | Одна тарелка |
|  | **m.** | один пирог | Один пирог |
|  | **n.** | один растение | Одно растение |
| **Polish** | **f.** | jeden kawa | Jedną kawa |
|  | **m.** | jeden człowiek | Jeden człowiek |
|  | **n.** | jeden mleko | Jedno mleko |

Table 4: Model inspection: examples

| **French** | **Norm.** | il vouloir d l amour d le joie de le bon humeur |
|---|---|---|
|  | **Out.** | Je veux d l amour d la joie de la bonne humeur |
| **Russian** | **Norm.** | девочка элиса жить в соседний подъезд |
|  | **Out.** | Девочка Элис живет в соседнем подъезде |
| **Polish** | **Norm.** | on dostać dużo oda nikt inny |
|  | **Out.** | Nie dostaniesz więcej od nikogo innego |

Despite being worse for short inputs, our model can still handles many important cases quite well including those discussed in Section 5.2.

## 5.4 DIALOG CORPUS

To evaluate our conversation model we constructed a corpus of question-answer pairs from web site `Otvet.mail.ru` — general topic Russian service for questions and answers (analogue of `Quora.com`). The uniqueness of this corpus is that it contains general knowledge questions that allow the trained model to answer questions about movies, capitals, etc. This requires many rare entity-related words to be in the vocabulary which makes it extremely large without normalization.

### Q-CONCORDE VS CONCORDE

First we compared Q-Concorde and Concorde models to show that Q-Concorde can indeed grasp important morphological features form a context. We also trained baseline models with context concatenated to input sentence (with a special delimiter in between). word and sentence accuracies are reported in Table 5. Again, Concorde model was able to outperform baselines even though it didn't have access to the context. Also, Q-Concorde model was able to improve Concorde's performance.

Table 5: Performance of Q-Concorde and baselines on `Otvet.mail.ru` dataset.

|  | **CharRNN** | | **Bigrams** | | **Hierarchical** | | **Concorde** | | **Q-Concorde** | |
|---|---|---|---|---|---|---|---|---|---|---|
|  | **w** | **s** | **w** | **s** | **w** | **s** | **w** | **s** | **w** | **s** |
| **train** | 72.95 | 38.75 | 47.43 | 34.81 | 74.69 | 35.36 | 81.61 | 44.43 | **83.04** | **48.24** |
| **test** | 72.83 | 38.88 | 48.58 | 36.63 | 74.99 | 35.2 | 81.51 | 43.75 | **83.13** | **48.10** |

We inspected cases on which Q-Concorde model showed better performance than Concorde (Table 6). In example 1 of this table, question was asked about a single object. Q-Concorde model used singular form, while Concorde used plural. Q-Concorde was also able to successfully carry correct case (example 2) and time (example 3) from the question.

Some mistakes made by Q-Concorde model are shown in Table 7. In example 1, Q-Concorde wasn't able to decide whether to use polite form or not and used one word in a less polite form than another.

An important property of Q-Concorde model is that it can generate different texts depending on lexical features of a question. For example, in Table 8 we changed question's tense from present simple to past simple ("what do you do?" and "what did you do?"). The model correctly generated

Table 6: Cases on which Q-Concorde is **better** than Concorde.

|   | | |
|---|---|---|
| **1** | **Question** | какой ты человек сложный или с тобой все просто |
| | **Concorde** | простые и добрые |
| | **Q-Concorde** | простой и добрый |
| **2** | **Question** | что прячут под вуалью |
| | **Concorde** | тайна |
| | **Q-Concorde** | тайну |
| **3** | **Question** | вас матери сколько носили месяцев |
| | **Concorde** | по стандарту выносят |
| | **Q-Concorde** | по стандарту выносили |

Table 7: Cases on which Q-Concorde is **worse** than Concorde.

|   | | |
|---|---|---|
| **1** | **Question** | какой тарифный план выбрать |
| | **Concorde** | позвоните вашему оператору |
| | **Q-Concorde** | позвони вашему оператору |
| **2** | **Question** | что у вас на завтрак |
| | **Concorde** | только поцелуи |
| | **Q-Concorde** | только поцелуй |

answer in corresponding tense. We also tried to change gender of a word "did" in a question: from masculine «делал» ) to feminine «далала» . Our model used the correct gender to generate the answer. While model generated grammatically correct answers in all three cases, in the third case (past simple, masculine) model answered in less common form with a meaning that differs from expected one.

Table 8: Inspection of Q-Concorde model.

| Question | Normalized answer | Answer |
|---|---|---|
| чего ты делаешь | | я хожу гулять |
| чего ты делала | я хожу гулять | я ходила гулять |
| чего ты делал | | я ходил гулял |

### 5.4.1 CONVERSATION MODEL

Finally, we apply Q-Concorde model to a proposed pipeline for training conversational models. We compare our model with a 3-layer character-level sequence-to-sequence model which was trained on grammatically correct sentences. For generating diverse answers we train two models: one to predict answer given question and the other to predict question given answer, as suggested in Li et al. (2015). This allows us to discard answers that are too general.

To compare two models we set an experiment environment where assessors were asked to select one of two possible answers to the given question: one was generated by character-wise model and another was generated by our pipeline. Assessors did not know the order in which cases were shown and so they did not know which model generated the text. In **62.1%** cases assessors selected the proposed model, and in the remaining **37.9%** assessors preferred character-wise model.

### 5.5 COMPUTATIONAL COSTS

We noticed that time for processing one batch is much higher for character-level models since they need to process longer sequences sequentially. In Table 9 we report time for forward and backward pass of one batch (16 objects) and other important computational characteristics. We measured this time on GeForce GTX TITAN X graphic card. It turns out that proposed models have comparable evaluation time, but train faster than unigram and hierarchical models.

Table 9: Computation performance of models.

| Model | Unigram | Hierarchical | Bigram | Concorde | Q-Concorde |
|---|---|---|---|---|---|
| Training time per batch | 410 ms | 379 ms | 284 ms | 190 ms | 285 ms |
| Number of parameters | 26M | 36M | 54M | 32M | 35M |
| Prediction time | 153 ms | 146 ms | 142 ms | 128 ms | 147 ms |
| GPU memory, train | 2316 Mb | 1970 Mb | 5602 Mb | 4290 Mb | 4512 Mb |
| GPU memory, evaluation | 1417 Mb | 2025 Mb | 2184 Mb | 3419 Mb | 4445 Mb |

## 6 DISCUSSION

Proposed model can be used beyond original neural conversation purposes. It can be used as a post-processing model for almost any text generation utils. For example, it can be used for snippet generation where multiple sentences come from different parts of a document and so may have morphological conflicts between each other. Sentences may even be taken from different articles like it happens in text generation for news services. Another interesting application of our model is predictive typing, where program predicts next word that user will type. While these systems often propose correct words, they usually suggest them in wrong forms.

## 7 CONCLUSION

In this paper we proposed a neural network model that can efficiently employ relationship between input and output words in morphological agreement task. We also proposed a modification for this model that uses context sentence. We apply this model for neural conversational model in a new pipeline: we use normalized question to generate normalized answer and then apply proposed model to obtain grammatically correct response. This model showed better performance than character level neural conversational model based on assessors responses.

We achieved significant improvement comparing to character-level, bigram and hierarchical sequence-to-sequence models on morphological agreement task for Russian, French and Polish languages. Trained models seem to understand main grammatical rules and notions such as tenses, cases and pluralities.

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
