# OpenReview forum: "Achieving morphological agreement with Concorde"
_ICLR.cc/2018/Conference — Reject_

### Official Review · AnonReviewer1 · 2017-11-24
**Unreadable paper**

**Rating:** 2
**Confidence:** 5

**Review:**

The paper is a pain to read. Most of the citation styles are off (i.e., without parentheses). Most of the sentences are not grammatically correct. Most, if not all, of the determiners are missing. It is ironic that the paper is proposing a model to generate grammatically correct sentences, while most of the sentences in the paper are not grammatically correct.

The experimental numbers look skeptical. For example, 1/3 of the training results are worse than the test results in Table 1. It also happens a few times in Table 5. Either the models are not properly trained, or the models are heavily tuned on the test set.

The running times in Table 9 are also skeptical. Why are the Concorde models faster than unigrams and bigrams? Maybe this can be attributed to the difference in the size of the vocabulary, but why is the unigram model slower than the bigram model?

---

### Official Review · AnonReviewer2 · 2017-11-28
**Borderline paper on morphological agreement**

**Rating:** 5
**Confidence:** 4

**Review:**

In this work, the authors propose a sequence-to-sequence architecture that learns a mapping from a normalized sentence to a grammatically correct sentence. The proposed technique is a simple modification to the standard encoder-decoder paradigm which makes it more efficient and better suited to this task. The authors evaluate their technique using three morphologically rich languages French, Polish and Russian and obtain promising results.

The morphological agreement task would be an interesting contribution of the paper, with wider potential. But one concern that I have is regarding the evaluation metrics used for it. Firstly, word accuracy rate doesn't seem appropriate, as it does not measure morphological agreement. Secondly, sentence accuracy (w.r.t. the sentences from which the normalized sentences are derived) is not indicative of morphological agreement: even "wrong" sentences in the output could be perfectly valid in terms of agreement. A grammatical error rate (fraction of grammatically wrong sentences produced) would probably be a better measure.

Another concern I have is regarding the quality of the baseline: Additional variants of the baseline models should be considered and the best one reported. Specifically, in the conversation task, have the authors considered switching the order of normalized answer and context in the input? Also, the word order of the normalized answer and/or context could be reversed (as is done in sequence-to-sequence translation models).

Also, many experimental details are missing from the draft:
-- What are the sizes of the train/test sets derived from the OpenSubtitles database?
-- Details of the validation sets used to tune the models.
-- In Section 5.4, no details of the question-answer corpus are provided. How many pairs were extracted? How many were used for training and testing?
-- In Section 5.4.1, how many assessors participated in the evaluation and how many questions were evaluated?
-- In some of the tables (e.g. 6, 7, 8) which show example sentences from Polish, Russian and French, please provide some more information in the accompanying text on how to interpret these examples (since most readers may not be familiar with these languages).

Pros:
-- Efficient model
-- Proposed architecture is general enough to be useful for other sequence-to-sequence problems

Cons:
-- Evaluation metrics for the morphological agreement task are unsatisfactory
-- It would appear that the baselines could be improved further using standard techniques

---

### Official Review · AnonReviewer3 · 2017-11-30
**Empirical results are convincing, contribution to representational learning is not much**

**Rating:** 6
**Confidence:** 5

**Review:**

The key contributions of this paper are:
(a) proposes to reduce the vocabulary size in large sequence to sequence mapping tasks (e.g., translation) by first mapping them into a "standard" form and then into their correct morphological form,
(b) they achieve this by clever use of character LSTM encoder / decoder that sandwiches a bidirectional LSTM which captures context,
(c) they demonstrate clear and substantial performance gains on the OpenSubtitle task, and
(d) they demonstrate clear and substantial performance gains on a dialog question answer task.

Their analysis in Section 5.3 shows one clear advantage of this model in the context of long sequences.

As an aside, the authors should correct the numbering of their Figures (there is no Figure 3) and provide better captions to the Tables so the results shown can easily understood at a glance.

The only drawback of the paper is that this does not advance representation learning per se though a nice application of current models.

---

### Decision · Program_Chairs · 2018-01-29
**ICLR 2018 Conference Acceptance Decision**

**Decision:**

Reject

**Comment:**

The pros and cons of this paper cited by the reviewers can be summarized below:

Pros:
* Empirical results demonstrate decent improvements over other reasonable models
* The method is well engineered to the task

Cons:
* The paper is difficult to read due to grammar and formatting issues
* Experiments are also lacking detail and potentially difficult to reproduce
* Some of the experimental results are suspect in that the train/test accuracy are basically the same. Usually we would expect train to be much better in highly parameterized neural models
* The content is somewhat specialized to a particular task in NLP, and perhaps of less interest to the ICLR audience as a whole (although I realize that ICLR is attempting to cast a broad net so this alone is not a reason for rejection of the paper)

In addition to the Cons cited by the reviewers above, I would also note that there is some relevant work on morphology in sequence-to-sequence models, e.g.:
* "What do Neural Machine Translation Models Learn about Morphology?" Belinkov et al. ACL 2017.

and that it is common in sequence-to-sequence models to use sub-word units, which allows for better handling of morphological phenomena:
* "Neural Machine Translation of Rare Words with Subword Units" Sennrich et al. ACL 2016.

While the paper is not without merit, given that the cons seem to significantly outweigh the pros, I don't think that it is worthy of publication at ICLR at this time, although submission to a future conference (perhaps NLP conference) seems warranted.